# Generation of hiPSC-Derived Skeletal Muscle Cells: Exploiting the Potential of Skeletal Muscle-Derived hiPSCs

**DOI:** 10.3390/biomedicines10051204

**Published:** 2022-05-23

**Authors:** Eric Metzler, Helena Escobar, Daniele Yumi Sunaga-Franze, Sascha Sauer, Sebastian Diecke, Simone Spuler

**Affiliations:** 1Max-Delbrück-Center for Molecular Medicine in the Helmholtz Association (MDC), Robert-Rössle-Str. 10, 13125 Berlin, Germany; helena.escobar@charite.de (H.E.); daniele.franze@mdc-berlin.de (D.Y.S.-F.); sascha.sauer@mdc-berlin.de (S.S.); sebastian.diecke@mdc-berlin.de (S.D.); 2Experimental and Clinical Research Center, a Cooperation between the Max-Delbrück-Center for Molecular Medicine in the Helmholtz Association and Charité—Universitätsmedizin Berlin, Lindenberger Weg 80, 13125 Berlin, Germany; 3Max-Delbrück-Center for Molecular Medicine in the Helmholtz Association (MDC), Genomics Platform, Hannoversche Straße 28, 10115 Berlin, Germany; 4Max-Delbrück-Center for Molecular Medicine in the Helmholtz Association (MDC), Pluripotent Stem Cells Platform, Robert-Rössle-Str. 10, 13125 Berlin, Germany; 5Charité—Universitätsmedizin Berlin, Corporate Member of Freie Universität Berlin and Humboldt-Universität zu Berlin, Experimental and Clinical Research Center, Lindenberger Weg 80, 13125 Berlin, Germany

**Keywords:** hiPSCs, muscle stem cells, reprogramming, epigenetic memory, myogenic differentiation, transplantation

## Abstract

Cell therapies for muscle wasting disorders are on the verge of becoming a realistic clinical perspective. Muscle precursor cells derived from human induced pluripotent stem cells (hiPSCs) represent the key to unrestricted cell numbers indispensable for the treatment of generalized muscle wasting such as cachexia or intensive care unit (ICU)-acquired weakness. We asked how the cell of origin influences efficacy and molecular properties of hiPSC-derived muscle progenitor cells. We generated hiPSCs from primary muscle stem cells and from peripheral blood mononuclear cells (PBMCs) of the same donors (*n* = 4) and compared their molecular profiles, myogenic differentiation potential, and ability to generate new muscle fibers in vivo. We show that reprogramming into hiPSCs from primary muscle stem cells was faster and 35 times more efficient than from blood cells. Global transcriptome comparison revealed significant differences, but differentiation into induced myogenic cells using a directed transgene-free approach could be achieved with muscle- and PBMC-derived hiPSCs, and both cell types generated new muscle fibers in vivo. Differences in myogenic differentiation efficiency were identified with hiPSCs generated from individual donors. The generation of muscle-stem-cell-derived hiPSCs is a fast and economic method to obtain unrestricted cell numbers for cell-based therapies in muscle wasting disorders, and in this aspect are superior to blood-derived hiPSCs.

## 1. Introduction

Cell replacement therapies for muscle wasting disorders are considered very promising therapeutic strategies [1,2]. Skeletal muscle disorders comprise of a large variety of acquired and genetic disorders, almost all of them characterized by muscle wasting and weakness. Muscular atrophy might be confined to certain muscle groups as in limb girdle muscular dystrophies (LGMD) or generalized as seen in cancer cachexia, intensive-care-unit-acquired weakness, or Duchenne’s muscular dystrophy. Skeletal muscle is not a transplantable organ such as the heart, kidney, liver, pancreas, or even skin. Therefore, therapeutic approaches focus on supporting muscle regeneration and restoration using external stimuli.

Physiologically, skeletal muscle regenerates through muscle-specific stem cells, so called satellite cells [3,4,5,6]. Muscle stem cells (MuSCs) cannot be replaced by other cell types such as mesenchymal stem cells, mesangioblasts, or by cytokines and growth hormones. Human satellite cells will have a place in regenerative medicine [7,8], but their relatively limited proliferative capacity will always remain a hindrance in repairing large muscle groups. Restoration of small, functionally crucial muscle has been demonstrated for swallowing muscles in oculopharyngeal muscular dystrophy (OPMD) [9] or urethral sphincter in congenital epispadias [10,11].

For the treatment of generalized muscle wasting disorders, cell replacement therapies must be developed from a source of unrestricted cell numbers. Human induced pluripotent stem cells (hiPSCs), differentiated into muscle progenitors, are such an option [2,12,13,14]. Indeed, several protocols have been established to differentiate them into muscle cells using the ectopic expression of myogenic transcription factors such as MYOD1 (Myogenic Differentiation factor 1) or PAX7 (Paired Box 7) [15,16] or directed transgene-free protocols recapitulating in vivo myogenesis to avoid genomic alterations [17,18,19,20,21,22]. Transgene-free protocols are based on Wnt (Wingless-INT) signaling [17,18,19,20,23], and some add BMP (Bone Morphogenetic Protein) inhibition to block lateral plate mesoderm induction and enhance differentiation towards the paraxial mesoderm [17,18,23]. Cells obtained with these protocols express key myogenic markers and spontaneously contract. Successful myofiber regeneration in vivo was shown [17,19,24] and additional cell purification approaches have recently been reported [25,26].

hiPSCs are most commonly generated from PBMCs or skin fibroblasts, but, theoretically, every somatic cell can be reprogrammed [27]. It is still a matter of debate to which degree the cell type of origin or the genetic background of the donor may influence the characteristics of the resultant hiPSCs. Donor differences have been described [28,29,30,31], as well as epigenetic differences between hiPSCs generated from different cell types of origin [32,33,34,35,36]. It has also been postulated that the cell type of origin may influence the redifferentiation potential of iPSCs with a preference to differentiate towards the lineage they originate from [37,38,39,40,41,42,43]. The findings are not consistent [29,44]. Probably, each cell type and differentiation pathway need to be addressed individually.

We dedifferentiated MuSCs and PBMCs from the same donors and compared their ability to generate hiPSCs, their transcriptome, and their potency to generate new muscle cells. Redifferentiated muscle cells were analyzed in vitro and in vivo after transplantation. MuSC-derived hiPSC colonies were obtained in significantly higher number and lower economic costs than PBMC-derived hiPSCs. The genetic signature of the resulting hiPSCs differed only in 122 transcripts, and both hiPSC cell types differentiated into induced myogenic cells with myofiber formation potential after transplantation. In order to establish hiPSC-derived tissue cells as an indispensable cornerstone of regenerative therapies, they must be efficacious, safe, and affordable. For the treatment of muscle wasting disorders with access to biopsy specimen, the possibility of generating unlimited myogenic cells from MuSC-derived hiPSCs could be of medical and economic advantage.

## 2. Materials and Methods

### 2.1. Donors

Muscle and blood samples were obtained from donors without histological signs of neuromuscular disorders in accordance with the requirements of the Charité Ethics Committee. Biopsies were obtained to exclude inflammatory disorders. Informed consent was obtained from all donors. Research use of the material was approved by the regulatory agencies (EA1/203/08, EA2/051/10, EA2/175/17, Charité Universitätsmedizin Berlin). Sex and age of each donor is listed in Table 1.

### 2.2. Primary MuSC Isolation and Culture

MuSC isolation was performed as described [7,8,45,46]. In brief: after the biopsy, the muscle biopsy specimen was transferred into Solution A and manually dissected using forceps under a stereomicroscope to obtain single muscle fiber fragments. Human muscle fiber fragments (HMFFs) were further cultured in Skeletal Muscle Cell Growth Medium (SMCGM, Provitro, Berlin, Germany) with myogenic cell populations growing out of the fragments. Primary MuSC populations used in this study are listed in Table 1. MuSCs were cultured in Skeletal Muscle Cell Growth Medium (SMCGM, Provitro, Berlin, Germany) at 37 °C and 5% CO_2_. MuSCs were detached using 0.25% trypsin-EDTA (Provitro, Berlin, Germany) or TrypLE Express (Gibco, Thermo Fisher Scientific, Waltham, MA, USA) at 37 °C for 5 min.

### 2.3. PBMC Isolation and Culture

Patient-derived full blood samples in EDTA tubes were used for sterile isolation of peripheral blood mononuclear cells (PBMCs). EDTA tubes were kept at room temperature (RT) and were processed within two hours after blood withdrawal. The full blood sample was transferred to a Vacutainer^®^ CPT™ mononuclear cell preparation tube (BD), inverted carefully 8–10 times, and centrifuged at RT for 20 min at 1850 g. The supernatant was mixed with 40 mL of 2 mM DPBS/EDTA, centrifuged for 10 min at RT at 500 g, and the resulting pellet resuspended in 4 mL 4 °C cold 2 mM DPBS/EDTA. Isolated PBMCs were cultured in StemPro™-34 SFM, StemPro™-34 Nutrient Supplement, L-Glutamine (2 mM), Stem Cell Factor (SCF) (100 ng/mL), FMS-Like Tyrosine Kinase 3 (FLT-3) (100 ng/mL), Interleukin 3 (IL-3) (20 ng/mL), Interleukin 6 (IL-6) (20 ng/mL), and Erythropoetin (Epo) (2 U/mL).

### 2.4. Reprogramming

All cell types were reprogrammed with the same technique using Sendai-virus delivery of OCT3/4, SOX2, KLF4, and c-MYC (CytoTune™-iPS 2.0 Sendai Reprogramming Kit, Invitrogen, Ref.: A16517). Nine to thirteen days after Sendai-virus infection cells were transferred to hypoxic conditions, as it was shown to enhance hiPSC generation [47]. For PBMC reprogramming of two donors (C + E) the protocol had to be adjusted several times and required mouse embryonic feeder cells (MEFs) instead of Matrigel as a substrate to obtain any hiPSC colonies. All cell lines used in this study were propagated to passage 13–15 times and fully characterized regarding hiPSC morphology, pluripotency marker expression (Immunofluorescence + Fluorescence activated cell sorting (FACS)), karyotype stability, absence of Sendai viruses, and formation of three germ layers [48,49]. Reprogramming efficiencies were determined counting the emerging hiPSC colonies in one 6-well plate for each reprogramming experiment after 30 days after Sendai-virus infection. The number of hiPSC colonies was then divided by the number of cells initially infected with Sendai viruses. The appearing iPSC colonies were identified by bright field microcopy and distinct iPSC morphology.

### 2.5. hiPSC Culture

hiPSCs were cultured in mTeSR™1 medium (Stemcell Technologies, Vancouver, BC, Canada) on Matrigel-coated (Corning™ hESC-qualified) 6-well plates. hiPSCs were cultured in hypoxic conditions at 5% O_2_ as this was shown to improve hiPSC quality [50,51]. Cells were passaged routinely at a ratio of 1:10 every 3 days using 0.5 mM PBS/EDTA. Mycoplasma was tested using the Venor^®^ GeM qOneStep kit (Minerva Biolabs, Berlin, Germany).

### 2.6. RNA Sequencing and Data Analysis

RNA from cultured hiPSCs was isolated using the NucleoSpin^®^ RNA/Protein isolation kit (Macherey-Nagel, Düren, Germany) and stored at −80 °C. RNA quantity was assessed using the Qubit^®^ 2.0 Fluorometer with the Qubit™ RNA HS Assay Kit (Thermo Fisher Scientific, Waltham, MA, USA) and RNA quality was analyzed by the 4200 TapeStation System together with the high sensitivity RNA ScreenTape (Agilent, Santa Clara, CA, USA). RIN values (RNA integrity number) were above 7 for all samples. Library preparation for total RNA-Sequencing was performed using the NEBNext Ultra II Directional RNA Library Prep Kit for Illumina^®^ (New England Biolabs, Ipswich, MA, USA) together with the NEBNext rRNA Depletion Kit (Human/Mouse/Rat) (New England Biolabs, Ipswich, MA, USA). An amount of 250 ng total RNA was used per sample following the manufacturer’s instructions. NEBNext^®^ Multiplex Oligos for Illumina^®^ (New England Biolabs, Ipswich, MA, USA) were used for indexing with a final PCR enrichment of adapter ligated DNA of 10 cycles. Purification of the final PCR reaction using Agencourt^®^ AMPure XP PCR cleanup beads (Beckman Coulter Life Sciences, Indianapolis, IN, USA) was performed twice to ensure a clean end product.

Library quality was checked with the 2100 Bioanalyzer together with the Bioanalyzer High Sensitivity DNA Analysis chips (Agilent, Santa Clara, CA, USA). Library quantity was determined using the Qubit™ dsDNA HS assay kit (Agilent, Santa Clara, CA, USA). Finally, samples were appropriately pooled with a final concentration of 10 mM per sample and sequenced with 2 × 76 + 16 In (paired end, 76 cycles, 16 indices) using the HiSeq 4000 System (llumina). For each sample, the transcript quantification at isoform level was estimated by RSEM (bowtie2 default parameters, GRCh38) [52]. DEseq2 paired designed and padj < 0.05 was used for the differential expression analysis between MuSCs vs. PBMC vs. MiPS vs. BiPS [53]. The differentially expressed isoforms were further annotated via Biomart R package (hsapiens_gene_ensembl). Pathway analysis and Gene Ontology (GO) term enrichment analysis were performed using ConsensusPathDB (http://cpdb.molgen.mpg.de/, accessed 15 December 2017–10 January 2018). Further analysis was performed using the Database for Annotation, Visualisation and Integrated Discovery (DAVID) (https://david.ncifcrf.gov, access date 10 January 2018).

### 2.7. Myogenic Differentiation

The myogenic differentiation, based on a previously described protocol [17,18], is based on Wnt pathway activation (CHIR) and BMP pathway inhibition (LDN). Differentiations were started in hiPSCs without morphological signs of spontaneous differentiation. MiPS and BiPS from the same donor were always differentiated simultaneously to exclude interexperimental differences. hiPSCs were seeded in different densities (2.5/3.5 × 10^4^ cells/cm^2^) on Matrigel-coated (Corning™ hESC-qualified) 6-well plates at 20% O_2_ in mTeSR™1 medium (Stemcell Technologies, Vancouver, BC, Canada) supplemented with 10 µM Rock inhibitor (Y-27632 2HCl, Biozol, Eching, Germany). The next day medium was changed to mTeSR1. After 8 h, differentiation was started by changing to 420 μL/cm^2^ DI-CL medium (DMEM/F12 (Thermo Fisher Scientific, Waltham, MA, USA), 1% Insulin-Transferrin-Selenium (ITS) (Thermo Fisher Scientific, Waltham, MA, USA), 1% Non-essential amino acids (NEAA) (Thermo Fisher Scientific, Waltham, MA, USA), 0.2% Penicillin-Streptomycin (Thermo Fisher Scientific, Waltham, MA, USA), 3 µM CHIR-99021 (Sigma-Aldrich, St. Louis, MO, USA), 0.5 µM LDN-193189 (Miltenyi Biotech, Bergisch Gladbach, Germany)). On day 3, 20 ng/mL Fibroblast growth factor (FGF) (R&D Systems, Minneapolis, MN, USA) was added to DI-CL medium. On day 6, medium was changed to DK-LHIF medium (DMEM/F12 (Thermo Fisher Scientific, Waltham, MA, USA), 15% Knockout™ Serum Replacement (Thermo Fisher Scientific, Waltham, MA, USA), 1% NEAA (Thermo Fisher Scientific, Waltham, MA, USA), 0.1 mM 2-Mercaptoethanol (Thermo Fisher Scientific, Waltham, MA, USA), 0.2% Penicillin-Streptomycin (Thermo Fisher Scientific, Waltham, MA, USA), 0.5 µM LDN-193189 (Miltenyi Biotech, Bergisch Gladbach, Germany), 10 ng/mL Hepatocyte growth factor (HGF) (R&D Systems, Minneapolis, MN, USA), 2 ng/mL Insulin-like growth factor 1 (IGF) (R&D Systems, Minneapolis, MN, USA), 20 ng/mL FGF (R&D Systems, Minneapolis, MN, USA)). On days 6–8 the different cell density approaches were evaluated and the technical triplicates of the seeding density with the highest cell survival were kept. On days 8–12 medium was changed to DK-I medium (DMEM/F12 (Thermo Fisher Scientific, Waltham, MA, USA)), 15% Knockout™ Serum Replacement (Thermo Fisher Scientific, Waltham, MA, USA), 1% NEAA (Thermo Fisher Scientific, Waltham, MA, USA), 0.1 mM 2-Mercaptoethanol (Thermo Fisher Scientific, Waltham, MA, USA), 0.2% Penicillin-Streptomycin (Thermo Fisher Scientific, Waltham, MA, USA), 2 ng/mL IGF (R&D Systems, Minneapolis, MN, USA)). On days 0–12 medium was changed every day. On day 12 10 ng/mL HGF (R&D Systems, Minneapolis, MN, USA) was added to DK-I medium and from now on medium was changed every other day. Day 30 cultures were enzymatically (TrypLE™ Express, Thermo Fisher Scientific, Waltham, MA, USA) and mechanically (25G needles) dissociated for a maximum of 45 min/6-well. Each enzymatical incubation did not exceed 10 min and cells were collected after each round of incubation in DMEM/F12 + 20% FBS. Single cell myogenic progenitor cells (iMPCs) were resuspended in skeletal muscle cell growth medium (SkGM-2 BulletKit, Lonza, Basel, Switzerland) + 10 μM Rock-inhibitor and seeded at a density of 6.5 × 10^4^ cells/cm^2^ on Matrigel-coated plates. Cells were propagated for 5 passages (15–20 days) and stocked in between. P5 cells were then differentiated into induced myogenic cells (iMCs) by incubation for 10 days in Terminal Differentiation medium (DMEM/F12 (Thermo Fisher Scientific, Waltham, MA, USA), 1% ITS (Thermo Fisher Scientific, Waltham, MA, USA), 0.2% Penicillin-Streptomycin (Thermo Fisher Scientific, Waltham, MA, USA), 1% L-Glutamin (Thermo Fisher Scientific, Waltham, MA, USA), 1% N2-Supplement (Thermo Fisher Scientific, Waltham, MA, USA)).

### 2.8. Immunofluorescence Staining in Cultured Cells

Induced myogenic cells (iMCs) grown on Matrigel-coated 8-well IbiTreat slides (Ibidi) were fixed with 3.7% Formaldehyde (10 min). Cells were permeabilized with 0.1% Tween 20 in TBS and blocked in 0.1% Triton X-100 + 1% FBS in TBS for 30 min. Primary antibodies were diluted in blocking solution and incubated overnight at 4 °C on a shaker (Desmin (1:1000, Abcam ab15200), fast MyHC (1:200, Sigma-Aldrich M4276), MyHC3 (1:200, Santa-Cruz sc-2064), Myogenin (1:800, Abcam ab1835), PAX7 (1:200, Santa-Cruz sc-81648), TUJ1 (1:1000, Sigma-Aldrich T8578)). Next day, secondary antibodies were also diluted in blocking solution and again incubated overnight at 4 °C on a shaker (AlexaFluor 568 goat anti-mouse (1:1000, Thermo Fisher A11031), AlexaFluor 568 donkey anti-rabbit (1:1000, Thermo Fisher A10042), AlexaFluor 488 goat anti-mouse (1:1000, Thermo Fisher A11001)). Hoechst (1:10,000 in DPBS) was incubated for 5 min at RT. Confocal immunofluorescence imaging was performed using the Laser Scan Microscope LSM 700 (Carl Zeiss, Jena, Germany) and for mosaic image acquisition a Leica DMI 6000 B microscope equipped with a XY scanning stage (Leica Microsystems) was used.

### 2.9. RT-qPCR

RNA from cultured cells was isolated using the NucleoSpin^®^ RNA/Protein isolation kit (Macherey-Nagel, Düren, Germany) and stored at −80 °C. An amount of 1 µg of isolated total RNA was reverse transcribed using the QuantiTect Reverse Transcription Kit (Qiagen, Venlo, The Netherlands) according to the manufacturer’s instructions. cDNA samples were stored at −20 °C. For PCR reaction, 10 μL KAPA SYBR^®^ FAST qPCR Master Mix (2X) Universal (Sigma-Aldrich, St. Louis, MO, USA) was mixed with 1 μL 5 μM primer mix (Forward + Reverse), 4 μL PCR grade water, and 5 μL of the diluted cDNA sample in the respective concentration to a final volume of 20 μL. Primer sequences are given in the Appendix A. Analysis was performed using a QuantStudio™ 6 Flex Real-Time PCR System (Thermo Fisher Scientific, Waltham, MA, USA) in three technical replicates per sample and primer pair. For primer pairs with optimal annealing temperature at 60 °C, a two-step protocol with 40 cycles with denaturation at 95 °C and annealing at 60 °C was used. For primer pairs with optimal annealing temperatures different from 60 °C, a three-step protocol was used with 40 cycles with denaturation at 95 °C, annealing (see Appendix A) and elongation and acquisition at 72 °C. Two control samples were used, one being an RT-qPCR reaction per primer without cDNA but water and the other an RT-qPCR reaction per primer using a reverse transcription product without reverse transcription enzyme.

Ct values were normalized to the housekeeping gene *GAPDH* (ΔCt values) for each sample, each primer pair, and each PCR plate. ΔΔCt values were calculated using the sample with the lowest expression as the relative expression value.

### 2.10. Mouse Strain and Irradiation of Hind Limbs

Mouse experiments were performed under the license number G0035/14. Immunodeficient, xenograft compatible, male NOG-M mice (NOD.Cg-*Prkdc^scid^II2rg^tm1Sug^*/JigTac, Taconic Biosciences, Bomholtvej, Denmark) were purchased at the age of 6–9 weeks, 1 week ahead of each experiment. Animals were kept in a specific-pathogen-free (SPF) animal facility with controlled temperature and humidity at the Max Delbrück Center for Molecular Medicine, Berlin, Germany. Focal irradiation of hind limb muscles, while avoiding irradiation of other body parts, was performed using an image-guided robotic system (CyberKnife Radiosurgery System, Accuray Inc., Sunnyvale, CA, USA) at the Charité CyberKnife facility (Virchow-Klinikum, Berlin, Germany) as previously described [7,54]. For irradiation, mice were anaesthetized with ketamine-xylazine in PBS (9 mg/mL ketamine, 1.2 mg/mL xylazine) with an intra peritoneal (i.p.) dose of 160 μL/20 g of body mass or by inhalation of 2% isoflurane. Digitally reconstructed radiographs (DDRs) were generated using a computer tomography (CT) scan with 0.75 mm slice thickness for radiation dose distribution, in order to reach the desired dose of 16 Gy in the three-dimensional target area of the limb.

### 2.11. Transplantation of iMPCs

For transplantation, freshly differentiated induced myogenic progenitor cells (iMPCs) were detached using TrypLE™ Express (Thermo Fisher Scientific, Waltham, MA, USA) and resuspended in DPBS + 2% FCS. Cells were stored on ice until transplantation. Mice were anaesthetized by inhalation of 2% isoflurane using an Univentor 400 anesthesia unit (Univentor, Zejtun, Malta) together with a flexible customized inhalation mask. When the anesthesia was effective, a specific area of the right hind limb, below the knee, above the tibialis anterior muscle was shaved and disinfected using isopropanol. An amount of 10^5^ cells in 11 μL injection volume were injected into the central part of the tibialis anterior muscle using a 25 μL model 702 RN SYR Hamilton^®^ syringe (Hamilton, Reno, NV, USA) coupled with a custom-made 20 mm long 26 G small hub removable needle. Mice were monitored on a daily basis and sacrificed 21 days after cell injection.

### 2.12. Histological Sections

Tibialis anterior (TA) muscles were dissected and cut in two halves following a transversal plane. Each TA half was separately mounted in gum tragacanth (12% gum tragacanth (Sigma-Aldrich, St. Louis, MO, USA), a few granules of Crystal Thymol (Synopharm, Beijing, China), and 4.6% glycerin in dH2O) on cork disks with the cutting surface facing the top. The embedded muscles were then frozen in chilled isopentane for 15 s and then immediately transferred to liquid nitrogen. For long-term storage, muscles were transferred to −80 °C. Muscles were sectioned using a Leica CM3050S Cryostat (Leica Biosystems, Nussloch, Germany) preparing 6 μm thick sections that were stored at −20 °C until further processing. 

### 2.13. Immunofluorescence Staining in Histological Tissue Sections

Sections were thawed and dried from −20 °C to RT for 45 min and then fixed for 5 min in −20 °C cold acetone. Sections were dried again for 10 min at RT and then blocked with 5% BSA + 3% goat serum in DPBS (Thermo Fisher Scientific, Waltham, MA, USA) for 1 h at RT in a humidity chamber. Sections were washed once with DPBS and then incubated with the primary antibody in 1% BSA in DPBS for 2 h at RT in a humidity chamber (human Spectrin (1:100, Novocastra NCL-SPEC1), human Lamin A/C (1:4000, Abcam ab108595)). Secondary antibodies were diluted in DPBS and incubated for 45 min at RT in a humidity chamber (Alexa Fluor 568 goat anti-rabbit (1:500, Thermo Fisher A11036), Alexa Fluor 647 goat anti-mouse (1:500, Thermo Fisher A21236)). Hoechst (1:5000 in DPBS) was incubated for 5 min at RT. Sections were mounted on microscope slides using Aqua-Poly/Mount (Polysciences, Inc.) and were left to dry overnight at 4 °C. Confocal immunofluorescence imaging was performed using the Laser Scan Microscope LSM 700 (Carl Zeiss).

### 2.14. Statistics

Statistical analysis was performed using the GraphPad Prism 8 software (GraphPad, San Diego, CA, USA). Differences were considered statistically significant for *p* < 0.05. Statistical tests performed for each specific experiment are indicated in the figure legends.

## 3. Results

### 3.1. MuSCs Reprogram more Efficiently into hiPSCs Than PBMCs

To perform a study free of interfering donor differences, we generated hiPSCs from primary human MuSCs (MiPS) and PBMCs (BiPS) from the same donor (Figure 1a; *n* = 5; Table 1). For reprogramming, standardized conditions based on Sendai-virus delivery of the OSKM (Oct4, Sox2, KLF4, c-Myc) reprogramming cocktail were used for both cell types (Figure 1b). The reprogramming efficiency using MuSCs as a source was 35× higher compared to PBMCs, with an average reprogramming efficiency of 0.07% for MuSCs and 0.002% for PBMCs (Figure 1c, Appendix A). Further, whereas MiPS generation was successful in the first attempt from all donors, multiple replications were necessary with two of five PBMC samples with one donor from whom no BiPS could be generated, despite multiple modifications of the reprogramming procedure (Appendix A). It should be mentioned that the reprogramming efficiency for MiPS from donor E (no BiPS) was considerably lower compared to the other donors, showing the influence of the genetic background of the donor on the ability to generate hiPSCs. However, in addition to the lack of BiPS from one donor, the multiple repetitions of the reprogramming procedures for PBMC samples resulted in considerable higher costs and expenditure of time using PBMCs compared to MuSCs as cells of origin to generate hiPSCs (Figure 1d).

### 3.2. Global Transcriptomic Analysis of the Generated MiPS and BiPS

Age- and sex-matched donors A, B, and D were included in the global transcriptome analysis (female, 47–50 years of age). Transcriptomic analysis showed a clear clustering of the MiPS and BiPS compared to their cell types of origin (Figure 2a). However, total RNA-Sequencing showed 122 significantly differentially regulated transcripts comparing MiPS and BiPS (adjusted *p* value *p* < 0.05) (Figure 2b). Gene Ontology (GO) term analysis with a significance threshold of *p* < 0.1 indicated biological processes involved in viral processes, cell–cell adhesion, DNA damage response, and chromatin remodeling. In addition, signaling pathways were found that are known to be involved in a wide range of regulatory signaling mechanisms, and more specifically also in differentiation processes such as Wnt, FGF, and TGF beta signaling (Figure 2c).

### 3.3. In Vitro Myogenic Differentiation Capacity of MiPS and BiPS

The myogenic differentiation capacities were compared using a transgene-free protocol (Figure 3a). This allowed a comparison without silencing intrinsic differentiation capabilities by overexpression of exogenic transcription factors. Immunofluorescence analysis showed the expression of Desmin and fast myosin heavy chain (fast MyHC) with no visual and quantifiable difference in the number of myogenic cells detected after differentiation (Figure 3b). Quantification of *n* = 9 independent experiments for three donors showed no significant difference in RNA expression levels for the myogenic genes *Desmin*, *PAX7*, *MYOD1*, *Myogenin,* and the myosin heavy chain isoforms *MYH2*, *MYH3,* and *MYH7* (Figure 3c, Appendix A). Non-myogenic markers such as the neuronal class III beta tubulin (TUJ1) and the platelet endothelial cell adhesion molecule-1 (*PECAM1*) were not detected in these cultures (Appendix A).

### 3.4. In Vivo Myofiber Formation Potential of Induced Myogenic Progenitor Cells (iMPCs) Generated from MiPS and BiPS

In order to test whether the myogenic cells generated from MiPS and BiPS can contribute to myofiber formation in vivo we transplanted single-cell-induced myogenic progenitor cells (iMPCs) from day 45 of the differentiation protocol into the irradiated tibialis anterior (TA) muscle of immunocompromised NOG-M mice (Figure 3a, Appendix A). In both MiPS-derived iMPC (M-iMPCs) and BiPS-derived iMPC (B-iMPCs) transplants, we detected human myofibers by human-specific anti-Spectrin antibody staining (Figure 4a, Appendix A). A quantification showed no difference in the number of donor-derived human myofibers comparing the two cell types (Figure 4b, Appendix A). However, the number of human myofibers in the graft (average 5) remained low compared to the transplantation of primary human MuSCs (average 20–30 [8]) and the generated human myofibers remained small (Appendix A). In addition, a high number of human nuclei, stained with a human-specific anti-Lamin A/C antibody, remained interstitial. PAX7-positive human nuclei were not detected in the grafts (Appendix A).

### 3.5. Analysis of Donor Background versus Cell Type of Origin

Interestingly, the global transcriptomic analysis showed clustering of MiPS and BiPS regarding the donor rather than the cell type of origin (Figure 5a). In addition, the in vitro myogenic differentiation comparison showed pronounced differences between the efficiencies of hiPSCs from different donors to differentiate into the myogenic lineage, as shown by immunofluorescence for the myogenic makers fast MyHC, MyHC3, Myogenin, Desmin, and PAX7 (Figure 5b). Supporting this, the proliferation rates of the isolated iMPCs also differ mainly regarding the donor (Appendix A). Interestingly, multinucleation, as a hallmark of MuSC fusion into myotubes, was detected in all donors, but nuclei in close proximity in a “pearl-chain”-like alignment were only detected in donor D. The muscle stem cell marker PAX7 was also detected in donor D iMCs only. In RT-qPCR analysis, donor D also differed significantly from donors A and B and therefore confirmed the immunofluorescence findings (Figure 5c, Appendix A).

## 4. Discussion

We show that primary MuSCs are superior to PBMCs as a source for hiPSCs in respect to time of generation and therefore costs. Further, muscle cells differentiated from hiPSCs have very similar properties independently of the origin of the hiPSC. Cell therapies in regenerative medicine are an important upcoming innovative approach for disorders such as muscle wasting. Success of individual products will depend on the quality of the produced cell as well as on their ability to compete in revenue and pricing.

The direct comparison of MuSCs and PBMCs for the generation of hiPSCs with cells from the same donors has not been performed before. However, it has previously been noted that MuSCs are indeed an efficient source of hiPSCs [55], while another study from the same group reported reprogramming of PBMCs with lower efficiencies [56]. We extended and improved the comparison in our study by using muscle cells and PBMCs from the same donor as well as by thoroughly defining the muscle stem populations that served as starting material.

Among the differentially regulated transcripts between MiPS and BiPS are gene products that are involved in the Wnt and FGF signaling pathways, essential regulators of differentiation. Yet, in our experimental setting the myogenic differentiation capacity of MiPS and BiPS was very similar. A reason might be that the Wnt and FGF pathway modification has been an intrinsic part of our myogenic differentiation protocol. Thus, the differences between MiPS and BiPS might have been masked by external addition of recombinant factors. Studies demonstrating an epigenetic memory of the cell of origin and an influence on the capacity to differentiate did not actively modify signaling pathways but analyzed spontaneous differentiation [37,38,57]. It could also be speculated that the effects of the intrinsic differences between hiPSCs may differ between the source and organ of the primary cell type, as many different cell types of origin and target tissues have been used in the reported studies [29,37,38,39,40,44]. Moreover, differences in iPSCs caused by the cell type of origin might be present in low passages [39] but may disappear with higher population doubling times. Last, we cannot rule out that our readouts—assessment of myogenic markers by immunofluorescence staining and quantitative RT-qPCR as well as functional capacity to build muscle after transplantation—were insufficient to detect minute differences between redifferentiated muscle cells.

Our experiments show that the donor background has a higher impact on the genetic variability and myogenic differentiation potential between hiPSCs than the cell type of origin, even in age- and sex-matched donor sets. This is in accordance with previous reports comparing fibroblast- and PBMC-derived hiPSCs in hematopoietic differentiation [29] and in hepatic differentiation [28]. Of note, we did not detect differences in the myogenic differentiation capacity between the hiPSC clones from the same donor.

The capacity of hiPSC-derived muscle cells to form myofibers in vivo demonstrates their potential to become applicable in a clinical context. The sensitivity of myofiber quantification, however, does not allow for the detection of differences between MiPS and BiPS from a single donor. Compared to transplanted primary MuSCs [7,8] the number of human myofibers in the transplant was lower and the diameter of the newly built muscle fibers was small. Moreover, many iMCs did not contribute to new fibers and remained interstitial, and the satellite cell niche was not repopulated by PAX7-positive cells. These limitations might be overcome in the future by further developed differentiation protocols. Reporter lines with fluorescently labeled endogenous PAX7 expression will allow to monitor and further modulate myogenic differentiation from hiPSCs [24,58].

Muscle biopsies for morphological and histochemical analysis of skeletal muscle have for many years been essential in the work-up of patients with suspected neuromuscular disorders. More recently, molecular genetics, in particular the advances of next-generation sequencing, have made diagnostic muscle biopsies somewhat dispensable. A muscle biopsy is an invasive procedure and benefits should be balanced carefully against potential concerns. We would, however, still consider the muscle biopsy as a valuable element in the care of a patient with inherited muscular dystrophy: 1. The myopathological analysis of diseased muscle contributes vastly to the understanding of pathophysiological concepts of a given disorder. 2. The possibility to isolate and propagate muscle stem cells allows to collect living cells for therapies [9,10], with CRISPR/Cas9-based genetic correction of muscular dystrophy causing mutations adding to the spectrum of potential therapeutic benefits [45,46]. 3. The rapid and efficient generation of hiPSCs, as described here, results in limitless material to test gene repair strategies and produce cells for regenerative medicine applications.

Although reprogramming of MuSCs is more efficient and cost effective than that of PBMCs, it is unlikely that MuSCs will surpass more easily obtainable cell types in the generation of hiPSCs. However, when muscle biopsies are being performed for diagnostic purposes, the purification of satellite cells should be considered. For ethical reasons, highly regenerative satellite cells, present in the biopsy specimen, should not be neglected when an invasive biopsy procedure is being performed.

## 5. Conclusions

hiPSCs can be generated highly efficiently from muscle stem cellsMuscle-derived hiPSCs save costs and time as compared to PBMC-derived hiPSCsMuscle-derived hiPSCs are yet to be improved as compared to primary human muscle stem cells

## Figures and Tables

**Figure 1 biomedicines-10-01204-f001:**
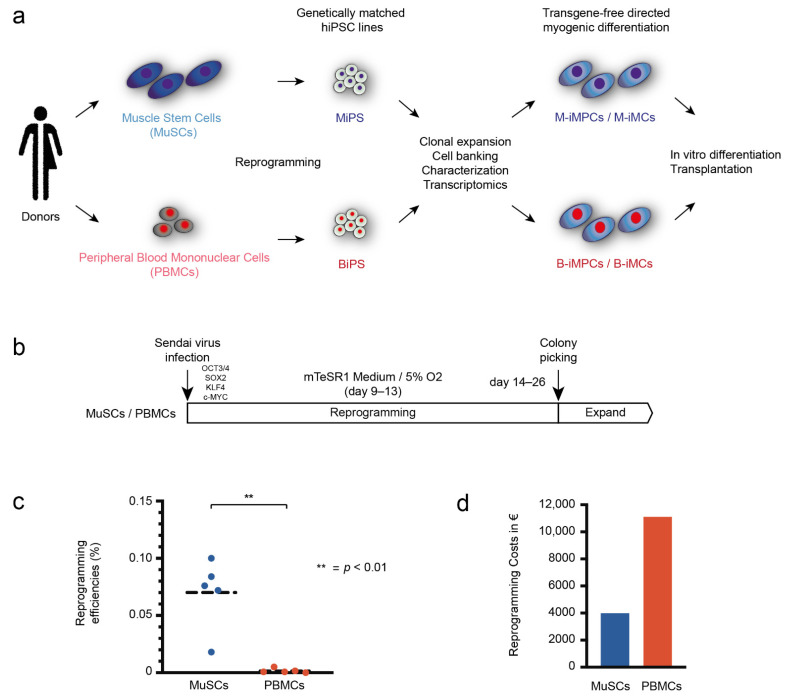
Efficient reprogramming of MuSCs into hiPSCs. (**a**) Schematic view of the study design. hiPSCs generated from MuSCs and PBMCs of the same donor are reprogrammed into hiPSCs (MiPS + BiPS), compared transcriptionally, and redifferentiated into induced myogenic cells (iMCs). MuSCs and PBMCs are shown in blue and red, respectively, throughout the figures. (**b**) All samples were reprogrammed using Sendai-virus delivery of the transcription factors OCT4, SOX2, NANOG, and c-MYC. Medium was changed to mTeSR1 medium after 9–13 days and cells were transferred to hypoxic conditions. (**c**) Quantification of the reprogramming efficiencies in percent comparing MuSCs and PBMCs (see also Appendix A). Statistics: Student’s *t*-test (*p* < 0.05). Dashed lines represent the mean. (**d**) Quantification of the costs for reprogramming of 5 MuSC and 5 PBMC samples in this study. All MiPS generations were successful with the first attempt; multiple repetitions were necessary with two of five PBMC samples, with one donor from whom no BiPS could be generated.

**Figure 2 biomedicines-10-01204-f002:**
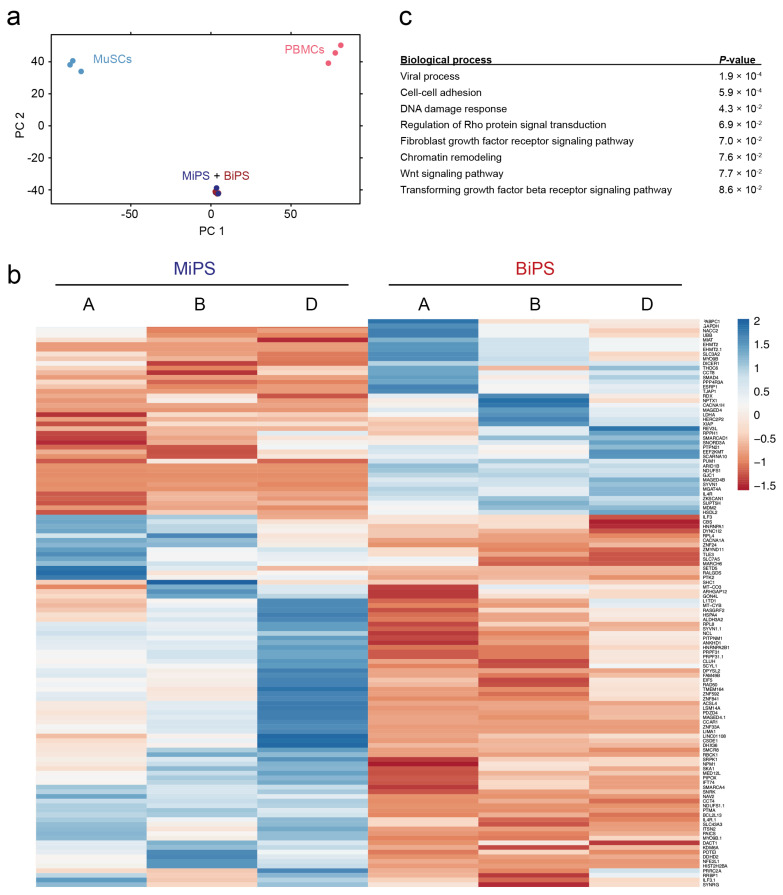
Global transcriptomic analysis comparing MiPS and BiPS. (**a**) Principal component analysis based on transcript expression level: MiPS (dark blue), MuSCs (light blue), BiPS (dark red), PBMCs (light red). hiPSCs of different origins cluster together with large differences compared to their cell type of origin. (**b**) Comparison of age- and sex-matched donors (female, 47–50 years of age; donors A, B, D). A total of 122 significantly differentially expressed transcripts (DETs) (padj < 0.05) comparing MiPS and BiPS, 80 upregulated in MiPS, 42 upregulated in BiPS. (**c**) DAVID GO term analysis for the 122 DET-enriched biological processes.

**Figure 3 biomedicines-10-01204-f003:**
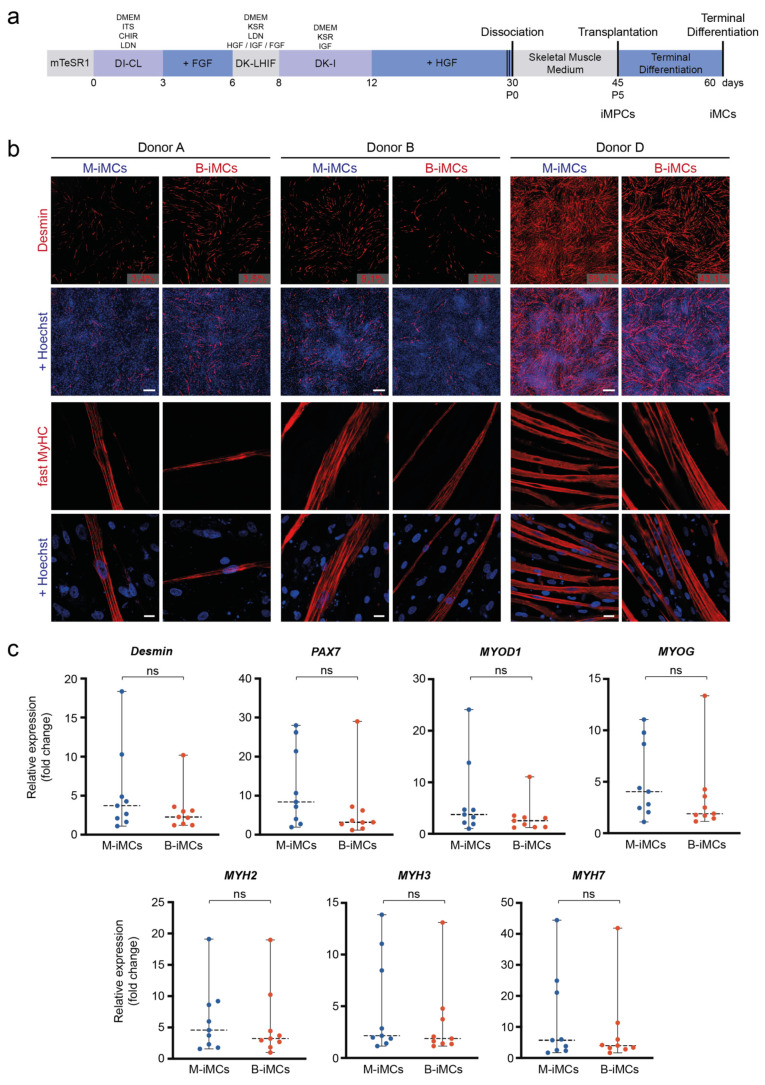
Comparable myogenic differentiation capacity for MiPS and BiPS. (**a**) Schematic description of the transgene-free myogenic differentiation protocol based on Wnt pathway activation and BMP pathway inhibition. iMCs: induced myogenic cells. iMPCs: induced myogenic progenitor cells. (**b**) Immunofluorescence imaging of iMCs differentiated from MiPS and BiPS after ~60 days of differentiation. Upper panel: Mosaic immunofluorescence imaging of iMCs stained with the myogenic marker Desmin. Percentage of Desmin-positive cells was calculated by manual counting >200 nuclei. Nuclei: Hoechst. Scale bar: 500 μm. Lower panel: Laser scanning microscopy of iMCs stained with the myogenic marker fast myosin heavy chain (MyHC). Nuclei: Hoechst. Scale bar: 20 μm. (**c**) RT-qPCR quantification of the myogenic markers *Desmin*, *PAX7*, *MYOD1*, *MYOG*, *MYH2/3/7* in iMCs from donors A, B, and D differentiated from MiPS and BiPS after ~60 days of differentiation. ΔΔCt values are shown as fold change relative to the sample with the lowest expression value: *n* = 9 independent differentiation experiments in each plot; *n* = 2–4 independent experiments for each donor. Each dot represents an independent experiment as mean of two technical replicates—details shown in Appendix A. In addition, Appendix A show ΔCt values. Statistics: Student’s *t*-test (*p* < 0.05); ns: not significant. Dashed lines represent the mean. Error bars in grey.

**Figure 4 biomedicines-10-01204-f004:**
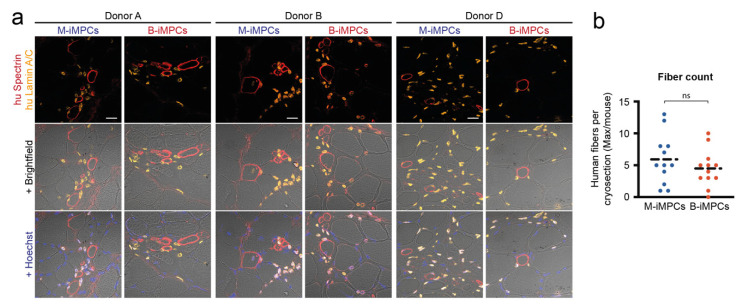
Human myofiber regeneration potential of M-iMPCs and B-iMPCs. Analysis of the cell transplantation experiment of M-iMPCs and B-iMPCs generated from donors A, B, and D. The hind limb of male 6–9-week-old xenograft compatible NOG-M mice (NOD.Cg-*Prkdc^scid^ II2rg^tm1Sug^*/JigTac) were irradiated with 16 Gray using an image-guided robotic radiosurgery system. Two days after irradiation, freshly differentiated and dissociated iMPCs (day 45 of differentiation) were injected into the central part of the tibialis anterior (TA) muscle. Then, 21 days after transplantation mice were sacrificed and TA muscles harvested. (**a**) Immunohistological analysis showing representative images of transversal TA muscle sections of grafted NOG-M mice. Sections are stained for human Spectrin, human Lamin A/C, and Hoechst. Scale bar: 20 μm. (**b**) Quantification of human muscle fibers in the transplants. The plotted values correspond to the section containing the highest number of human myofibers in each transplant. Statistics: Student’s *t*-test (*p* < 0.05); ns: not significant. Dashed lines represent the mean.

**Figure 5 biomedicines-10-01204-f005:**
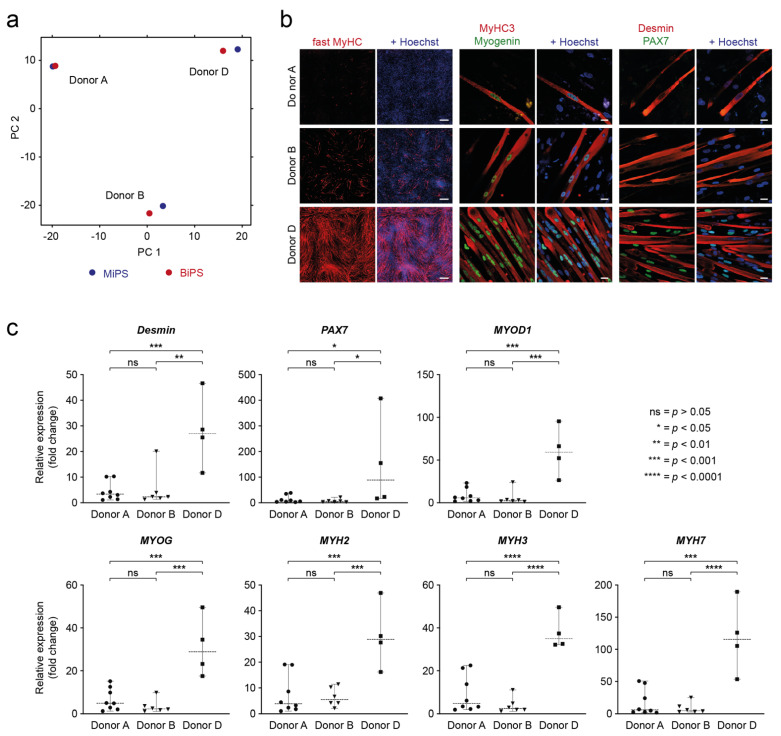
Influence of donor background on myogenic differentiation potential. (**a**) Principal component analysis based on transcript expression level showing MiPS and BiPS of 3 age- and sex-matched donors A, B, and D. MiPS (dark blue) and BiPS (dark red). Donor samples cluster together. (**b**) Immunofluorescence imaging of iMCs differentiated from donors A, B, and D after ~60 days of differentiation. Left panel: Mosaic immunofluorescence imaging of iMCs stained with the myogenic marker fast myosin heavy chain (MyHC). Nuclei: Hoechst. Scale bar: 500 μm. Middle and right panel: Laser scanning microscopy of iMCs co-stained with myogenic markers myosin heavy chain 3 (MyHC3) + Myogenin and Desmin + Pax7. Nuclei: Hoechst. Scale bar: 20 μm. (**c**) RT-qPCR quantification of myogenic marker expression for *Desmin*, *PAX7*, *MYOD1*, *MYOG*, *MYH2/3/7* in iMCs comparing donors A, B, and D. Graphs present the same data as shown in Figure 3C without separation for MiPS and BiPS. ΔΔCt values are shown as fold change relative to the sample with the lowest expression value. Appendix A shows the replicates that have been transplanted in the in vivo experiments. Appendix A shows ΔCt values. Donor A: *n* = 8; donor B: *n* = 6; donor D: *n* = 4. Each dot represents an independent experiment as mean of two technical replicates. Statistics: one-way ANOVA (*p* < 0.05). Dashed lines represent the mean. Error bars in grey.

**Table 1 biomedicines-10-01204-t001:** Samples used in this study.

**Donor**	Sex/Age	Cell Type	Cell Line (hPSCreg)	Cell Type of Origin	Passage	Successful Reprogramming	RNA-Seq	In Vitro Myogenic Differentiation	Transplantation
A	female, 47	MuSCs				yes	yes		
		PBMCs				yes	yes		
		hiPSCs	MDCi011-A	MuSCs	13		yes	yes	yes
		hiPSCs	MDCi011-B	PBMCs	15		yes	yes	yes
B	female, 50	MuSCs				yes	yes		
		PBMCs				yes	yes		
		hiPSCs	MDCi012-A	MuSCs	15		yes	yes	yes
		hiPSCs	MDCi012-B	PBMCs	15		yes	yes	yes
C	male, 18	MuSCs				yes			
		PBMCs				yes			
D	female, 47	MuSCs				yes	yes		
		PBMCs				yes	yes		
		hiPSCs	MDCi013-A	MuSCs	13		yes	yes	yes
		hiPSCs	MDCi013-B	PBMCs	15		yes	yes	yes
E	male, 58	MuSCs				yes			
		PBMCs				no			

## Data Availability

The RNA-Seq data presented in this study are openly available in the NCBI Sequencing Read Archive (SRA) at http://www.ncbi.nlm.nih.gov/bioproject/826262, BioProject ID PRJNA826262, Release date: 13 April 2022.

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
