# Peer review of "Generation of hiPSC-Derived Skeletal Muscle Cells: Exploiting the Potential of Skeletal Muscle-Derived hiPSCs"

_biomedicines, 2022, doi:10.3390/biomedicines10051204_

Round 1
Reviewer 1 Report
The work of Metzler at al., is well designed and comprehensive. The authors used PBMC and muscle stem cells from 4 donors to create hiPSCs, which were subsequently differentiated toward skeletal muscle cells in vitro and in vivo. The authors found that it was easier to create iPSCs from muscle stem cells and that these iPSCs do not differ much from PBMC-derived iPSCs. No differences in the differentiation capacity of the iPSCs, derived from PBMC and muscle progenitor cells, toward skeletal muscle cells were found. The differences in the differentiation toward skeletal muscle cells was more donor-dependent. The manuscript is well structured and easy to follow.
However, there are a few issues, which the authors need to clarify:
- Was the differentiation efficiency toward muscle cells similar between the experiments with the same donor line?
- After the differentiation toward muscle cells it is clear that there are also undifferentiated cells (shown in the pictures) , therefore the expression level of muscle-specific genes is highly reliable to the presence/absence of these undifferentiated cells in the culture. Did the authors observe significant interexperimental variability? Could the authors speculate what is the phenotype of the undifferentiated cells and is there a way to purify the muscle progenitor cells for more repetitive differentiation results? The lack of PECAM expression (as shown in the supplementary data) should be presented as delta Ct as this expression could not be detected in other lines and it is not clear what was the sample with the lowest expression.
- From the in vitro data it is clear that the cells from donor D differentiate more efficiently to muscle cells in comparison to the other two donors, whereas in vivo this difference is no longer visible. How the authors can explain that?
- Before the in vivo experiment did the authors somehow compare the purity of the injected progenitor cells? Did the authors expect differences in the progenitor potential of the cells between the different donors, which seems to be the case at least from the in vitro experiments? Could the expression of progenitor markers in the iMPCs be compared to the primary donor-derived muscle stem cells?
- It is not really clear how the authors calculated the reprogramming efficiency. In M&M section the authors wrote: “Reprogramming efficiencies were determined counting the emerging hiPSC colonies in one 6-well for each reprogramming experiment after 30 days after Sendai-virus infection.” How the reprogramming percentage was calculated? How the authors determined whether the appearing colonies were iPSCs?
- In Fig. 1b is shown that the reprogramming (days 9-13) was performed in hypoxic condition. Please explain why. This was not required in the Cytotune kit. The hypoxic environment is not routinely used for PBMC reprogramming and thus this information should be also included in the M&M section.
- For the RT-PCR results calculations, I would suggest to use the delta Ct method instead on ddCt method. This would give better understanding of the expression levels in the different samples.
- Could the authors clarify what was the phenotype of the muscle stem cells (as the authors refer to) used for reprogramming? In the manuscript of Marg at al, 2014, cited by the authors, the population obtained from muscle biopsy after 3 weeks of culture starts to loose Pax7 considered as marker of the muscle progenitor cells. At which timepoint the muscle stem cells were used for reprogramming and was this identical between the donors?
Author Response
Response to Reviewer 1 Comments:
We thank the reviewer for the time and comments.
Point 1: Was the differentiation efficiency toward muscle cells similar between the experiments with the same donor line?
Response 1: Supporting information figure S1 shows the variation in between the experiments for each cell line. We noticed variation in the differentiation efficiency from experiment to experiment highlighting the need for robust differentiation protocols. However, MiPs and BiPs from the same donor were always differentiated in parallel to exclude differences from experiment to experiment. We added that information to the methods (lines 185/186).
Point 2: After the differentiation toward muscle cells it is clear that there are also undifferentiated cells (shown in the pictures), therefore the expression level of muscle-specific genes is highly reliable to the presence/absence of these undifferentiated cells in the culture. Did the authors observe significant interexperimental variability? Could the authors speculate what is the phenotype of the undifferentiated cells and is there a way to purify the muscle progenitor cells for more repetitive differentiation results? The lack of PECAM expression (as shown in the supplementary data) should be presented as delta Ct as this expression could not be detected in other lines and it is not clear what was the sample with the lowest expression.
Response 2: We noticed interexperimental variability in the myogenic differentiation efficiency for each hiPSC line as shown in supporting information figure S1 and explained under response 1.
The non-myogenic cells differ in their identity. Our preliminary experiments showed a heterogenous expression of the mesodermal marker TBX6 on day 4 of the differentiating culture indicating cells of ectodermal or endodermal origin. To exclude at least some ectodermal or endodermal terminal differentiated cells we confirmed the absence of TUJ1 positive cells for ectodermal and PECAM1 positive cells for endodermal origin.
Purification of the myogenic progenitor population using FACS sorting was reported [25,26]. In our study, we determined the efficiency of myogenic differentiation of MiPs and BiPs. This question does not require FACS sorting.
The experiment was performed by RT-qPCR and for all differentiated samples the melt curve analysis did not show any PCR product for PECAM1. For HUVECs the melt curve showed a well-defined PCR product. Therefore, we decided not to show any CT values for the differentiated samples as the CT values have no meaning without a PCR product. We added that information to the legend of supporting information figure S4 (Supporting information, lines 57/58).
Point 3: From the in vitro data it is clear that the cells from donor D differentiate more efficiently to muscle cells in comparison to the other two donors, whereas in vivo this difference is no longer visible. How the authors can explain that?
Response 3: The overall myofiber formation efficiency, represented by the number of myofibers formed by human cells after transplantation of iMCs, is low in comparison to the transplantation of primary human muscle stem cells (Marg et al., 2014/2019). The sensitivity of myofiber quantification, however, does not allow for the detection of differences between MiPS and BiPS from a single donor. We added that information to the discussion (lines 509-511).
Point 4: Before the in vivo experiment did the authors somehow compare the purity of the injected progenitor cells? Did the authors expect differences in the progenitor potential of the cells between the different donors, which seems to be the case at least from the in vitro experiments? Could the expression of progenitor markers in the iMPCs be compared to the primary donor-derived muscle stem cells?
Response 4: The myogenic marker Desmin was 54% in iMPCs, PAX7 was 6%, and Ki67 was 81%. The quantification was performed in exemplary samples. In comparison, primary MuSC populations are 100% Desmin positive, and the number of nuclei stained positive for PAX7 ranges from 0-98% (Marg et al., 2014/2019). hiPSC-derived iMCs and primary MuSCs therefore differ in their degree of purity. The percentage of cells positive for PAX7 is with 6% rather low in the iMPC population but we recently showed that primary MuSCs can regenerate muscle also independent of PAX7 (Marg et al. 2019).
However, we did expect differences in between the donors as we detected differences in the number of myogenic cells generated from each donor (Figure 3/ Figure 5).
Point 5: It is not really clear how the authors calculated the reprogramming efficiency. In M&M section the authors wrote: “Reprogramming efficiencies were determined counting the emerging hiPSC colonies in one 6-well for each reprogramming experiment after 30 days after Sendai-virus infection.” How the reprogramming percentage was calculated? How the authors determined whether the appearing colonies were iPSCs?
Response 5: To calculate the reprogramming efficiencies the number of emerging hiPSC colonies was divided by the number of cells initially infected with Sendai viruses. The appearing iPSC colonies were identified by bright field microcopy and distinct iPSC morphology. We added that information to the methods section (lines 142-145).
Point 6: In Fig. 1b is shown that the reprogramming (days 9-13) was performed in hypoxic condition. Please explain why. This was not required in the Cytotune kit. The hypoxic environment is not routinely used for PBMC reprogramming and thus this information should be also included in the M&M section.
Response 6: It has been described by others that stemness and quality of iPSCs were improved when cultured under hypoxic conditions (Thomson et al. 2020; Kuijk et al. 2020). Furthermore, it was shown that hypoxic conditions enhance the generation of iPSCs (Yoshida et al. 2009). Thus, we performed both, reprogramming and maintenance hiPSC culture, under hypoxic conditions. We added the citations to the methods (lines 133/134 and 148/149).
Point 7: For the RT-PCR results calculations, I would suggest to use the delta Ct method instead on ddCt method. This would give better understanding of the expression levels in the different samples.
Response 7: In addition to the ddCt data we added supporting information in figures S2, S3 and S7 showing all RT-qPCR data with the dCt method.
Point 8: Could the authors clarify what was the phenotype of the muscle stem cells (as the authors refer to) used for reprogramming? In the manuscript of Marg at al, 2014, cited by the authors, the population obtained from muscle biopsy after 3 weeks of culture starts to loose Pax7 considered as marker of the muscle progenitor cells. At which timepoint the muscle stem cells were used for reprogramming and was this identical between the donors?
Response 8: The myoblasts used for reprogramming were passage 5 or younger (not longer than 2 weeks in cell culture) before infection with Sendai virus. The cell population was characterized by staining for Desmin, PAX7 and MYOD1.
Reviewer 2 Report
In this manuscript, Metzler et al. compare primary muscle stem cells (MuSCs) and peripheral blood mononuclear cells (PBMCs) as a source of hiPSCs for the generation of large quantities of skeletal muscle cells. They show that MuSCs can be reprogrammed into hiPSCs with significantly higher efficiency than PBMCs, reducing the associated costs. However, there was no noticeable difference in the myogenic differentiation potential of MuSC-derived hiPSCs compared to PBMC-derived hiPSCs. Donor background had the greatest impact on genetic variability and myogenic differentiation potential, rather than the source of the hiPSCs.
The study is interesting and relevant. The introduction contextualizes the work and refers potential applications of the object of research. The materials and methods are suitably described and the results are discussed objectively. The combination of transcriptomic analysis with in vitro and in vivo assays comprises a thorough and holistic approach that allows for robust conclusions.
Overall, the writing quality is good and the text is well organized, allowing for a prompt assimilation of the information and ideas being conveyed. A small number of minor flaws (e.g. missing commas, repeated words, etc.) do exist. Their correction is not imperative for comprehending the work, but could be the subject of improvement by conducting an additional grammatical revision.
Despite the high quality of the work, there are a few points that should be addressed before it can be considered for publication in Biomedicines. Below are some comments to help the authors improve their manuscript.
Major Points:
- It is not common practice to routinely culture hiPSCs in hypoxic conditions (5% O2). Is the information provided on line 126 correct, or is it mistakenly referring to the hiPSC reprogramming conditions, which usually include maintaining cells under hypoxia? If the information is correct, the use of hypoxia should be justified;
- In section 3.3, when referring to Figure 3b (lines 332 to 335), it is mentioned that no visual difference in the number of myogenic cells is detected after differentiation of MiPS and BiPS. Whenever possible, a purely visual inspection should be avoided, since it provides considerably less reliable conclusions than quantitative measurements. Could a quantitative measurement be implemented based on the immunofluorescence imaging? If it could, it would positively enhance the presented results;
- The main hypothesis of this work was that MuSCs would potentially be a better source of hiPSC-derived skeletal muscle cells than PBMCs given that MiPS might have improved myogenic potential comparatively to BiPS. However, this was not found to be the case based on the performed experiments. On the other hand, MuSCs were shown to have a 35x higher reprogramming efficiency than PBMCs, thus being significantly superior for hiPSC generation in terms of cost and time. Was this expected? Is there previously published regarding the reprogramming efficiency of MuSCs? If this is a novel result, it should be further highlighted as such, and if not, it should be further contextualized in the existing literature;
- Apart from the references, a brief description of the methodology used for MuSC harvesting should be included in the main text (in section 2.2) to contextualize readers who are not familiar with the protocol and to compare with PBMC isolation.
- Following up on the previous points, a more in-depth discussion of the advantages and disadvantages of using MuSCs versus PBMCs could enrich the Discussion section of this work. Will MuSCs ever come to rival or even surpass PBMCs as a source of hiPSCs? Do the reduced reprogramming costs of MuSCs seem to outweigh the more invasive nature of muscle biopsies? These questions are of particular interest because PBMCs, along with fibroblasts, are currently one of the most common sources of hiPSCs, while MuSCs are not typically considered for this purpose.
- All abbreviations should be defined when they are first used. As an example, in the Abstract, “hiPSCs” is defined (line 21), but “ICU” (line 23) and “PBMCs” (line 25) are not. A thorough revision should be undertaken to correct all such cases;
- Nomenclature should be as consistent as possible to avoid unnecessary confusion. The terms “hiPSCs” and “hiPS cells” are employed interchangeably, but it would be better to use only one of these, preferably “hiPSCs”;
Minor Points:
- The spacing between values and units is sometimes suppressed and should thus be carefully revised. “5min” (line 102) and “4mL” (line 110) are examples of this;
- In lines 82 and 88, the appropriate adjective is “economic” and not “economical”;
- In line 100, the company’s name is “Provitro”, as written in the next line;
- In line 105, a more appropriate verb would be “processed” instead of “proceeded”;
- In line 142, a better verb is “ensure” instead of “assure”;
- Revise the sentence starting with “On days” in line 172;
- In line 178, the determiner “every” should be used in place of the adverb “very”;
- In line 210, an erroneous line break was introduced in the middle of a sentence;
- In lines 269, 271 and 273, “humid chamber” should be replaced with “humidity chamber”;
- In line 284, “efficient” should be exchanged for “efficiently”;
- Figure 1b: in the schematic and its respective caption 4 transcription factors are mentioned, namely OCT4, SOX2, NANOG and c-MYC, but previously it was mentioned that KLF4 was used for reprogramming cell into hiPSCs instead of NANOG;
- In line 305, the medium is called “mTeSR1” and not “mTeST1”.
Author Response
Response to Reviewer 2 Comments:
We thank the reviewer for the time and comments.
Major Points:
Point 1: It is not common practice to routinely culture hiPSCs in hypoxic conditions (5% O2). Is the information provided on line 126 correct, or is it mistakenly referring to the hiPSC reprogramming conditions, which usually include maintaining cells under hypoxia? If the information is correct, the use of hypoxia should be justified;
Response 1: It has been described by others that stemness and quality of iPSCs were improved when cultured under hypoxic conditions (Thomson et al. 2020; Kuijk et al. 2020). Furthermore, it was shown that hypoxic conditions enhance the generation of iPSCs (Yoshida et al. 2009). Thus, we performed both, reprogramming and maintenance hiPSC culture, under hypoxic conditions. We added the citations to the methods (lines 133/134 and 148/149).
Point 2: In section 3.3, when referring to Figure 3b (lines 332 to 335), it is mentioned that no visual difference in the number of myogenic cells is detected after differentiation of MiPS and BiPS. Whenever possible, a purely visual inspection should be avoided, since it provides considerably less reliable conclusions than quantitative measurements. Could a quantitative measurement be implemented based on the immunofluorescence imaging? If it could, it would positively enhance the presented results;
Response 2: In addition to visual assessment, we quantified the number of Desmin positive cells and added the information to the figure 3b and text (line 375 and 393). In addition, RT-qPCR quantification was used (Fig. 3c).
Point 3: The main hypothesis of this work was that MuSCs would potentially be a better source of hiPSC-derived skeletal muscle cells than PBMCs given that MiPS might have improved myogenic potential comparatively to BiPS. However, this was not found to be the case based on the performed experiments. On the other hand, MuSCs were shown to have a 35x higher reprogramming efficiency than PBMCs, thus being significantly superior for hiPSC generation in terms of cost and time. Was this expected? Is there previously published regarding the reprogramming efficiency of MuSCs? If this is a novel result, it should be further highlighted as such, and if not, it should be further contextualized in the existing literature;
Response 3: A comparison between muscle cells and PBMCs in terms of reprogramming efficiency has been performed before by Trokovic et al. (2013, 2014 [55,56]). In the publication from 2013, myoblasts were reported to be reprogrammed with an efficiency of 0.75%. The study from 2014 reported PBMC reprogramming with an efficiency of 0.005%. We extended and improved the comparison in our study by using muscle cells and PBMCs from the same donor as well as by thoroughly defining the muscle stem populations that served as starting material. The confirmation of the Trokovic studies did not come as a surprise to us.
We added a comment in the discussion (lines 480-484).
Point 4: Apart from the references, a brief description of the methodology used for MuSC harvesting should be included in the main text (in section 2.2) to contextualize readers who are not familiar with the protocol and to compare with PBMC isolation.
Response 4: We added a summary of the MuSC isolation method (lines 110-115).
Point 5: Following up on the previous points, a more in-depth discussion of the advantages and disadvantages of using MuSCs versus PBMCs could enrich the Discussion section of this work. Will MuSCs ever come to rival or even surpass PBMCs as a source of hiPSCs? Do the reduced reprogramming costs of MuSCs seem to outweigh the more invasive nature of muscle biopsies? These questions are of particular interest because PBMCs, along with fibroblasts, are currently one of the most common sources of hiPSCs, while MuSCs are not typically considered for this purpose.
Response 5: We added to the discussion: Although reprogramming of MuSCs is more efficient and cost effective than that of PBMC it is unlikely that MuSCs will surpass more easily obtainable cell types in the generation of hiPSCs. However, when muscle biopsies are being performed for diagnostic purposes the purification of satellite cells should be considered. For ethical reasons, highly regenerative satellite cells, present in the biopsy specimen, should not be neglected when an invasive biopsy procedure is being performed (lines 533-538).
Point 6: All abbreviations should be defined when they are first used. As an example, in the Abstract, “hiPSCs” is defined (line 21), but “ICU” (line 23) and “PBMCs” (line 25) are not. A thorough revision should be undertaken to correct all such cases;
Response 6: Abbreviations have been reviewed and definitions added at first appearance.
Point 7: Nomenclature should be as consistent as possible to avoid unnecessary confusion. The terms “hiPSCs” and “hiPS cells” are employed interchangeably, but it would be better to use only one of these, preferably “hiPSCs”;
Response 7: We changed the whole manuscript to the term hiPSCs.
Minor Points:
Point 1: The spacing between values and units is sometimes suppressed and should thus be carefully revised. “5min” (line 102) and “4mL” (line 110) are examples of this;
Response 1: Manuscript has been revised.
Point 2: In lines 82 and 88, the appropriate adjective is “economic” and not “economical”;
Response 2: We changed the term to “economic”.
Point 3: In line 100, the company’s name is “Provitro”, as written in the next line;
Response 3: The “-“ has been deleted.
Point 4: In line 105, a more appropriate verb would be “processed” instead of “proceeded”;
Response 4: We changed the term to “processed”.
Point 5: In line 142, a better verb is “ensure” instead of “assure”;
Response 5: We changed the term to “ensure”.
Point 6: Revise the sentence starting with “On days” in line 172;
Response 6: Sentence has been revised.
Point 7: In line 178, the determiner “every” should be used in place of the adverb “very”;
Response 7: We changed the term to “every”.
Point 8: In line 210, an erroneous line break was introduced in the middle of a sentence;
Response 8: The break was eliminated.
Point 9: In lines 269, 271 and 273, “humid chamber” should be replaced with “humidity chamber”;
Response 9: We changed the term to “humidity chamber”.
Point 10: In line 284, “efficient” should be exchanged for “efficiently”;
Response 10: We changed the term to “efficiently”.
Point 11: Figure 1b: in the schematic and its respective caption 4 transcription factors are mentioned, namely OCT4, SOX2, NANOG and c-MYC, but previously it was mentioned that KLF4 was used for reprogramming cell into hiPSCs instead of NANOG;
Response 11: NANOG has been replaced with KLF4 in figure 1b.
Point 12: In line 305, the medium is called “mTeSR1” and not “mTeST1”.
Response 12: We changed the term to “mTeSR1”.
Reviewer 3 Report
See file enclosed

Author Response
Response to Reviewer 3 Comments:
We thank the reviewer for the time and comments.
Major revisions:
Point 1: The properties of skeletal muscle cells are quite different regarding donor’s origin. Can a statistical relationship be established between graft capacity, cell growth rate and muscle marker expression? It seems that the donor’s effect is dominant over the other factors. However, this result might be different with increasing donor numbers and should also be developed. Have you tried differentiating and transplanting iPSCs from male C and E donors? These experiments will consolidate the article’s conclusions.
Response 1: As always, when investigating human samples, interindividual differences are to be expected and need to be considered when drawing conclusions. We have generated hiPSCs from donors C and E but as it was not possible to obtain BiPs from donor E we decided to perform the differentiation analysis with a sex and age matched donor set. Therefore, we did not differentiated cells from donor C and E into iMCs.
Point 2: For the transplantation experiment, details of the injected control cells (noticed in the legend of figure S3) and the results obtained with these cells should be added. Moreover, for each donor, are the transplanted cells issued from the same differentiation assay? This needs to be clarified since several differentiation experiments were performed on cells of each donor (4 for A, 3 for B and 2 for C).
Response 2: We added information about the control cells to the text and referred to our previous publication that reported very successful engraftment of primary human muscle stem cells (Marg et al., 2019) (lines 412-414).
Our work focuses on the comparison between MiPS and BiPS. Thus, each differentiation experiment for MiPS and BiPS from the same donor was performed simultaneously on parallel plates using the same freshly prepared medium. The differentiation experiments from different donors were not handled simultaneously. We added the information about the replicate that was transplanted to supporting information figure S1 and added another supporting information figure S7.
Point 3: Finally, data about the functionality of the graft should be added. Did the lesion prevent the mice from walking? Did the cell transplantation solve this problem?
Response 3: The transplantation into NSG mice that do not carry a neuromuscular defect were designed to study donor cell engraftment, contribution to host muscle fibers and repopulation of the stem cell niche. Clinical improvement can only be measured in a disease model.
Minor revisions:
Point 4: The article presents 5 figures and 3 additional figures, 1 table and 2 additional tables. The numbering of these figures and tables should be checked and placed in the correct order of citation in the text (consider especially the Figures S2 and S3).
Response 4: We changed the internal numbering of figure S3 (now figure S5). We changed figure S2c (now figure S4) to figure S6. We exchanged table S1 and table S2.
Point 5: The age of the donor C is different in Table 1 and S1.
Response 5: The age has been corrected.
Point 6: Surprisingly, the efficacy of iPSC reprogramming is lower for one of the two male donors (Table S1), from PBMCs as described in the results (lines 291-298), but also from MuSCs: this difference may be commented.
Response 6: We added a comment on this difference (lines 327-330).
Point 7: In the same way, explanations of the transcriptomic differences between MiPSCs and BiPSCs should be more fully developed in the discussion (Lines 415-422), particularly regarding the variation also observed between donors.
Response 7: We focused our work on the comparison between MiPS and BiPS and their differentiation capacity into induced myogenic cells. However, the RNA-Seq dataset was uploaded and further analysis about donor differences may be performed: http://www.ncbi.nlm.nih.gov/bioproject/826262.
Round 2
Reviewer 3 Report
Thank you for your answers to my concerns, your article is much clearer with the changes made.